Subject Category:
Biology (whole organism)

Subject Areas:
ecology/behaviour

Keywords:
African wolf, ecosystem services, Ethiopian highlands, Ethiopian wolf, feeding ecology, pest rodents

Author for correspondence:
Nils C. Stenseth
e-mail: n.c.stenseth@ibv.uio.no

# Foraging ecology of African wolves (*Canis lupaster*) and its implications for the conservation of Ethiopian wolves (*Canis simensis*)

Tariku Mekonnen Gutema[1,3], Anagaw Atickem[4,5], Diress Tsegaye[2], Afework Bekele[5], Claudio Sillero-Zubiri[6,7], Jorgelina Marino[6,7], Mohammed Kasso[5], Vivek V. Venkataraman[8], Peter J. Fashing[1,9] and Nils C. Stenseth[1,5]

[1]Centre for Ecological and Evolutionary Synthesis (CEES), Department of Biosciences, and [2]Department of Biosciences, University of Oslo, PO Box 1066, Blindern, 0316 Oslo, Norway
[3]Department of Natural Resources Management, Jimma University, PO Box 307, Ethiopia
[4]Cognitive Ethology Laboratory, German Primate Center, Leibniz Institute for Primate Research, Kellnerweg 4, 37077 Göttingen, Germany
[5]Department of Zoological Sciences, Addis Ababa University, PO Box 1176, Addis Ababa, Ethiopia
[6]Wildlife Conservation Research Unit, Zoology Department, University of Oxford, Tubney House, Tubney OX13 5QL, UK
[7]IUCN SSC Canid Specialist Group, Oxford, UK
[8]Institute for Advanced Study in Toulouse, Toulouse 31000, France
[9]Department of Anthropology and Environmental Studies Program, California State University Fullerton, 800 North State College Boulevard, Fullerton, CA 92834, USA

VVV, 0000-0001-5016-4423; NCS, 0000-0002-1591-5399

African wolves (AWs) are sympatric with endangered Ethiopian wolves (EWs) in parts of their range. Scat analyses have suggested a dietary overlap between AWs and EWs, raising the potential for exploitative competition, and a possible conservation threat to EWs. However, in contrast to that of the well-studied EW, the foraging ecology of AWs remains poorly characterized. Accordingly, we studied the foraging ecology of radio-collared AWs ($n = 11$ individuals) at two localities with varying levels of anthropogenic disturbance in the Ethiopian Highlands, the Guassa-Menz Community Conservation Area (GMCCA) and Borena-Saynt

National Park (BSNP), accumulating 845 h of focal observation across 2952 feeding events. We also monitored rodent abundance and rodent trapping activity by local farmers who experience conflict with AWs. The AW diet consisted largely of rodents (22.0%), insects (24.8%), and goats and sheep (24.3%). Of the total rodents captured by farmers using local traps during peak barley production (July to November) in GMCCA, averaging $24.7 \pm 8.5$ rodents/hectare/day, 81% ($N = 3009$) were scavenged by AWs. Further, of all the rodents consumed by AWs, most (74%) were carcasses. These results reveal complex interactions between AWs and local farmers, and highlight the scavenging niche occupied by AWs in anthropogenically altered landscapes in contrast to the active hunting exhibited by EWs in more intact habitats. While AWs cause economic damage to local farmers through livestock predation, they appear to play an important role in scavenging pest rodents among farmlands, a pattern of behaviour which likely mitigates direct and indirect competition with EWs. We suggest two routes to promote the coexistence of AWs and EWs in the Ethiopian highlands: local education efforts highlighting the complex role AWs play in highland ecosystems to reduce their persecution, and enforced protection of intact habitats to preserve habitat preferred by EWs.

## 1. Introduction

The midsize canids in northern Africa considered to be golden jackals (*Canis aureus*) were recently reclassified as African wolves (AWs), *Canis lupaster*, due to their close phylogenetic relationship to the grey wolf (*C. lupus*) [1,2]. AWs are found throughout the Ethiopian Highlands, often in sympatry with endangered Ethiopian wolves (EWs), *Canis simensis* [3,4], Africa's most threatened carnivore. At fewer than 500 individuals, the EW is the rarest canid in the world [5]. Restricted to several enclaves of Afro-alpine habitats, small EW populations are highly vulnerable to extinction, particularly because of habitat loss as well as rabies and canine distemper virus outbreaks stemming from interactions with local domestic animals [6,7].

Recent scat analyses revealed that the diet of AWs consists largely of rodents (48–57%) and varies by season [8,9]. Given that EWs depend on abundant rodent populations for their survival and reproduction [10–12], potential niche overlap and competition between these two species might have negative effects on EW populations. Based on intensive study at multiple sites, EWs are known to be active rodent hunters and only rarely kill livestock or scavenge [10,11]. However, because our knowledge of the diet of AWs in the Ethiopian Highlands is based primarily on scat analyses [8,9], we do not know the proportion of rodents acquired through hunting versus scavenging rodents killed by local farmers using traditional practices. To better understand the nature and extent of potential competition between EWs and AWs, it is necessary to learn more about the AW's diet. If AWs primarily scavenge rodents, direct exploitative competition between these species may be relatively minor. On the other hand, if AWs primarily engage in the active hunting of rodents, particularly where the two species overlap, the potential for competition may be significant. Recent work has indicated that sympatric AWs and EWs do actively defend their territories from each other via agonistic interactions [8].

A better characterization of AW foraging ecology will also permit inferences about the nature of human-wildlife conflict in the Ethiopian Highlands. AWs are presently considered one of the main livestock predators in the Ethiopian Highlands and are heavily persecuted [9]. However, they may also provide an ecological benefit to farmers if they feed upon pests such as rodents and insects, which cause significant damage to crops in small-holder farms in Ethiopia [13–16].

Accordingly, our goal is to evaluate the foraging ecology of AWs in greater detail than before and to assess the potential effects their dietary choices may have on EWs via competition for resources. We intensively studied the foraging ecology of the AW in the Ethiopian Highlands via direct observations of 11 radio-collared individuals at two sites, and compared our results with those from published studies of the diet and foraging behaviour of EWs. Specifically, we estimated (1) the proportion of rodents in the diet of the AW that derived from scavenging versus predation, (2) the extent to which AWs foraged in farmland versus intact habitat and (3) rodent abundance and level of trapping activity by local farmers who experience conflict with AWs due to sheep predation.

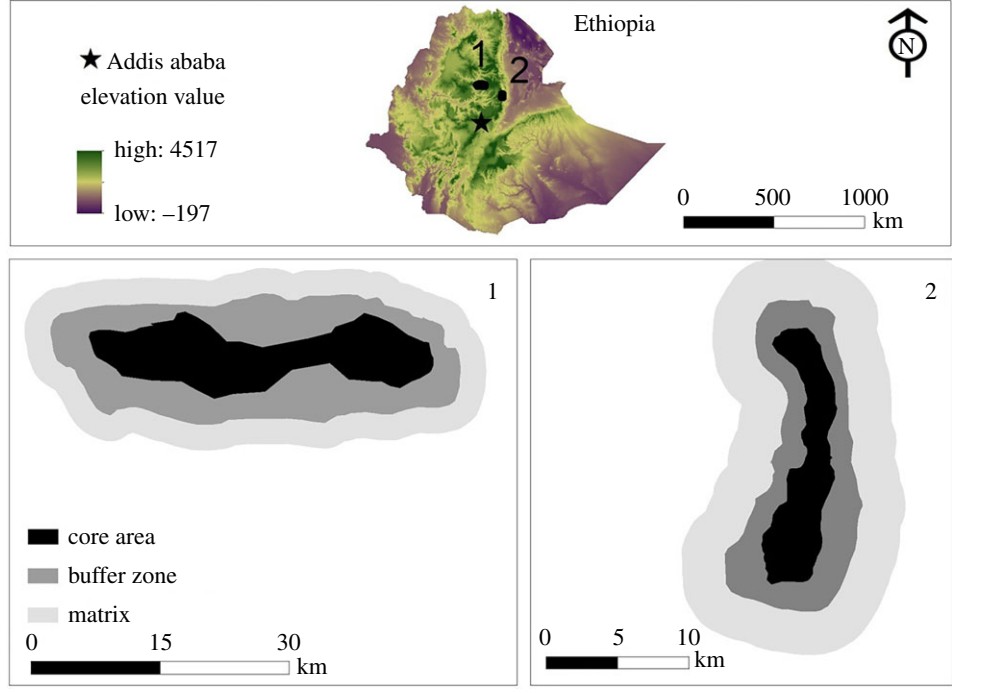

**Figure 1.** Study localities, (1) Borena-Saynt National Park (BSNP) and (2) Guassa-Menz Community Conservation Area (GMCCA).

## 2. Methods

### 2.1. Study area

Our study was carried out in Guassa-Menz Community Conservation Area (GMCCA; 10°15′–10°27′ N to 39°45′–39°49′ E) and Borena-Saynt National Park (BSNP; 10°50′–10°53′ N to 38°40′–38°54′ E; figure 1), areas of Afro-alpine habitat located in the north-central highlands of Ethiopia. GMCCA spans 111 km$^2$ with an elevational range of 3200–3600 m [17], while BSNP covers an area of 153 km$^2$ with an elevational range of 1900–3700 m [18,19]. Both sites are also home to several mammal species endemic to the Ethiopian Highlands, including EWs, gelada monkeys (*Theropithecus gelada*) and Starck's hare (*Lepus starcki*) [11,18]. The Ethiopian wolf populations are estimated at approximately 21 individuals in GMCCA [11] and approximately 16 individuals in BSNP [5]. The local people in both areas are mostly agro-pastoralists who grow barley between June and November and keep a variety of livestock (mostly sheep but also goats, cattle and horses) [8,18]. The two study areas are 150 km apart, but their climates are broadly similar, with a wet season extending from June to November and a dry season from December to May [8,11,19]. Detailed climatic data are available only for GMCCA where rainfall averages 1650 ± 243 mm per year, average monthly temperature is 11.0 ± 1.2°C, and mean monthly low and high temperatures are 4.3 ± 0.5°C and 17.8 ± 0.3°C, respectively (*n* = 6 years) [17].

Livestock grazing is a common practice in most Ethiopian protected areas [20,21]. Based on the levels of anthropogenic disturbance, we divided each study area into three zones: *core* (the section of the protected areas where all human and livestock activities are prohibited), *buffer* (the section of the protected areas where controlled livestock grazing is permitted), and *matrix* (human-dominated areas adjacent to the protected area which consist mainly of farmland and settlements [8]). In GMCCA, EWs largely occupy the core while AWs mostly use the buffer zone [8].

### 2.2. Foraging behaviour

We captured 11 AWs using rubber-lined Soft Catch foothold traps (Woodstream Corporation, Lititz, Pennsylvania, USA) sizes 1.5 and 3 (for method details see [19]) and fitted them with VHF radio collars; two males and three females in GMCCA and three males and three females in BSNP (for more details see [19]). In the wet and dry seasons of 2016 and 2017, we followed collared individuals for 3–4 days per month during both day and night. Focal observations were carried out with

binoculars from distances of 50–150 m after locating the focal individuals using a hand-held directional antenna. We recorded their activity, including successful and unsuccessful feeding attempts. A successful attempt was scored if the prey was killed and ingested. Accordingly, an unsuccessful attempt was scored when they failed to capture and kill the prey [22]. Scavenging was defined as feeding on a dead animal, typically taking dead rodents from traps set by farmers (*difit*, see below) ([10,11]; electronic supplementary material, table S1). We classified the food items consumed by AWs as hunted prey (including common molerats *Tachyoryctes splendens*, smaller rodents, and shrews), livestock carcasses (cattle, horses, sheep and goats), rodent carcasses (taken from *difit* traps) and arthropods (mainly grasshoppers, but also spiders and beetles). Whenever we observed AWs feeding or attempting to capture prey, we recorded the appropriate habitat type, classified as bushland (greater than 50% shrubs, predominantly *Helichrysum* and *Erica* spp.), grassland (greater than 50% open land; including rocky grassland, open grazing land dominated by *Festuca* spp.) or farmland (barley and other crops).

## 2.3. Traditional traps '*difit*' as a source of rodents for AWs

In both GMCCA and BSNP, farmers use a traditional trapping method known as *difit* to protect their crops from rodents [9]. However, for this specific objective, we collected data only from GMCCA. *Difit* are made of a locally manufactured rope, a relatively heavy stone and some barley seeds as bait (electronic supplementary material, figure S1). We collected data on the use of *difit* in GMCCA, where 25 barley farm sites adjacent to GMCCA were investigated, recording the number of rodents trapped (per hectare per day) and the extent to which AWs exploited the traps by taking the dead rodents. Farmers usually set up their traps in the morning (07.00–9.00 h), visiting them at 1–4 h intervals and resetting them if a capture had taken place. Trapping concluded in the evening (17.00–18.00 h). Every morning, after the traps were set we checked them regularly at 2 h intervals. When we found rodents had been caught in the trap, we recorded their number (usually one per trap, but occasionally two) and the species to which they belonged. We then cleared and reset the trap. In addition, we recorded whenever carnivores and raptors were observed taking rodents from the traps.

## 2.4. Data analysis

We compared the proportions of food items consumed by the AWs in the GMCCA and BSNP by a mixed effect model using food items as response variable, localities as fixed effects and individual collared animals as random effects.

We estimated successful hunting by AWs on two food classes (rodents and sheep) in relation to seasons, using logistic regression by fitting a general linear model. The response variable was binomial (1/0), indicating successful or unsuccessful hunting attempts, respectively. The fixed effects were diet (at two levels: rodents and sheep) and season (at two levels: dry and wet). We combined the data collected from GMCCA and BSNP (proportion of food items consumed) after verifying no significant differences for the two sites using one-way ANOVA followed by Tukeys's HSD post hoc test.

We also compared the effect of habitat types (i.e. fixed effect factor at three levels: bushland, farmland and grassland) on the success probability (attempt to feed and outcome) of AWs capturing rodents (i.e. binomial response variable: 1/0 where 1 is successful) using logistic regression.

All analyses were done in R v. 3.3.1 (R Core Team 2016).

# 3. Results

## 3.1. Foraging ecology observations

We observed radio-tracked AWs for 845 h across 16 months during 2016 and 2017 (392 h in GMCCA and 453 h in BSNP) resulting in a total of 2952 records of food items. Our focal observations revealed that the proportion of food items consumed by AWs in both study areas did not differ (electronic supplementary material, table S3). We found that AWs consumed primarily rodents during the wet season, and ate a more diverse diet, including more insects, livestock carcasses and sheep, during the dry season (figure 2 and table 1). Indeed, dietary composition differed significantly between the wet and dry seasons: rodents ($z = 94.6$, d.f. = 1, $p < 0.001$), sheep ($z = 22.4$, d.f. = 1, $p < 0.001$) and insects ($z = 38.6$, d.f. = 1, $p < 0.001$). The probability of AWs successfully hunting rodents (successful events as a

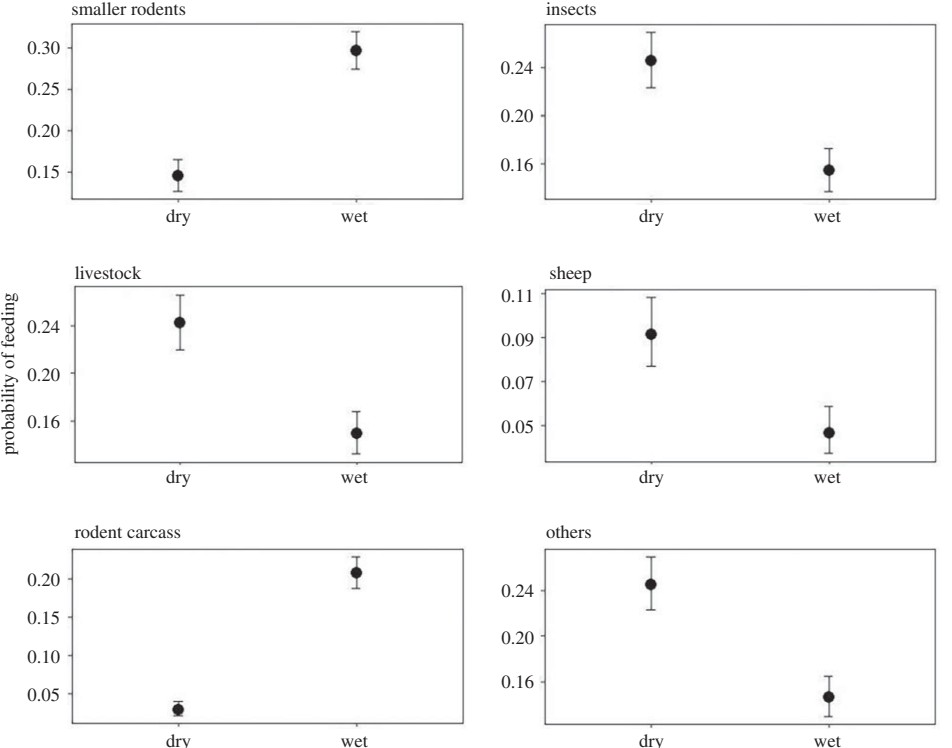

**Figure 2.** Probability of African wolves feeding on different diets in the dry and wet seasons.

**Table 1.** Composition of African wolf diet. Recorded as successful hunting attempts from focal animal observations of 11 individuals in GMCCA and BSNP.

| food items | total $n = 2952$ | BSNP | | | GMCCA | | |
| --- | --- | --- | --- | --- | --- | --- | --- |
| | | dry $n = 902$ | wet $n = 753$ | total $n = 1655$ | dry $n = 450$ | wet $n = 847$ | total $n = 1297$ |
| small rodents | 22.76 | 16.30 | 31.08 | 23.02 | 11.56 | 28.22 | 22.44 |
| arthropods | 19.00 | 26.72 | 17.00 | 22.30 | 18.44 | 12.87 | 14.80 |
| livestock caracasses | 18.56 | 20.51 | 14.48 | 17.76 | 29.78 | 14.17 | 19.58 |
| unidentified | 18.53 | 23.28 | 16.73 | 20.30 | 25.11 | 11.57 | 16.27 |
| rodent carcasses | 12.13 | 2.22 | 14.74 | 7.92 | 4.00 | 24.68 | 17.50 |
| sheep | 6.50 | 9.20 | 2.92 | 6.34 | 8.22 | 5.90 | 6.71 |
| grass | 1.32 | 0.78 | 1.86 | 1.27 | 1.11 | 1.53 | 1.39 |
| potatoes | 0.54 | 0.22 | 0.93 | 0.54 | 0.22 | 0.71 | 0.54 |
| wild birds | 0.20 | 0.33 | 0.13 | 0.24 | 0.44 | 0.00 | 0.15 |
| duikers | 0.17 | 0.33 | 0.00 | 0.18 | 0.22 | 0.12 | 0.15 |
| chickens | 0.14 | 0.11 | 0.00 | 0.06 | 0.44 | 0.12 | 0.23 |
| hares | 0.14 | 0.00 | 0.13 | 0.06 | 0.44 | 0.12 | 0.23 |

proportion of total hunting attempts) also differed between seasons ($z = 4.6$, d.f. = 1, $p < 0.001$), but not on sheep ($z = 1.5$, d.f. = 1, $p = 0.2$; figure 3).

Among the 491 rodents consumed by AWs, 28% were hunted and 72% were scavenged from traps. AWs exhibited a higher proportion of successful feeding attempts in farmland (36%, $n = 229$) than in

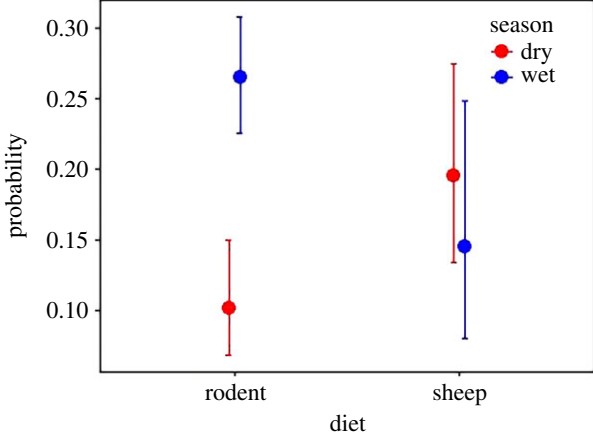

**Figure 3.** Probability of African wolves successfully capturing (successful events per total hunting attempts) rodents and sheep during hunting attempts in the wet and dry seasons.

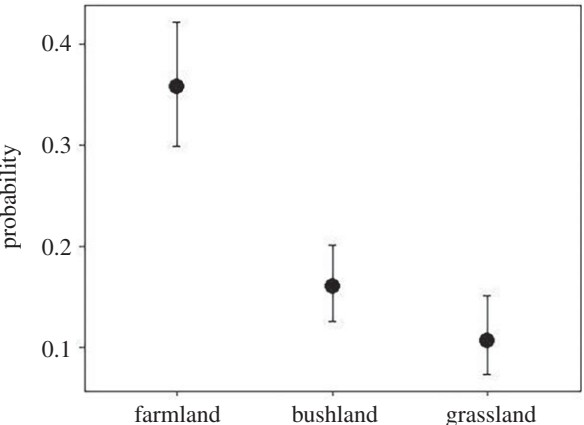

**Figure 4.** Probability of African wolves successfully capturing rodents in different habitat types.

**Table 2.** Comparison of African wolves' success in capturing rodents (active hunting) in different habitat types using Tukey multiple comparisons test.

| habitat | different | lower | upper | *p* adj |
|---|---|---|---|---|
| farmland–bushland | 0.197 | 0.121 | 0.274 | 0.0000 |
| grassland–bushland | −0.054 | −0.129 | 0.021 | 0.212 |
| grassland–farmland | −0.251 | −0.335 | −0.168 | 0.0000 |

bushland (17.3%, $n = 324$) or grassland (10.7%, $n = 244$) (figure 4). The proportion of successful feeding attempts did not differ between bushland and grassland (table 2).

During the study period, we observed AWs killing 192 sheep, of which 163 (85%) were killed by solitary AWs, 21 (11%) by pairs and four (2%) by groups of three AWs.

## 3.2. AW foraging on dead rodents from traps

During the period of barley production (July–November) in 2016 and 2017, 3009 rodents were trapped in *difits* at an average rate of $24.7 \pm 8.5$ rodents $\mathrm{ha}^{-1}\,\mathrm{d}^{-1}$ (electronic supplementary material, table S2). The Natal multimammate mouse (*Mastomys natalensis*) was the most frequently captured species in *difits*, accounting for 72.6% of the total. The other four captured species were the Ethiopian white-footed mouse (*Stenocephalemys albipes*, 17.3%), grey-tailed narrow-headed rat (*S. griseicauda*, 5.1%), Abyssinian grass rat (*Arvicanthis abyssinicus*, 4.3%) and *Lophuromys* spp. (0.8%). Eleven rodent species were caught

**Table 3.** Frequency (%) of rodents and shrews ($n = 420$) captured in the three zones of GMCCA using Sherman live traps and percentage of rodent species captured in farmland using *difit*. For comparison, frequency (%) of occurrence of rodents per scat (348 scat samples) of EWs in the same study site (data from [11]).

| species | matrix | | buffer | | core | | total | captured in *difit* | EW |
|---|---|---|---|---|---|---|---|---|---|
| | dry | wet | dry | wet | dry | wet | | | |
| Arvicanthis abyssinicus | 0.2 | 0.0 | 0.0 | 0.0 | 0.0 | 0.0 | 0.2 | 4.3 | 59.5 |
| Dendromus lovati | 0.0 | 0.0 | 0.0 | 0.0 | 0.2 | 0.0 | 0.2 | 0 | |
| Lophuromys brevicaudus | 0.0 | 1.0 | 4.5 | 14.5 | 12.4 | 12.9 | 45.2 | 0.8 | |
| Lophuromys flavopunctatus | 0.0 | 0.0 | 1.4 | 1.4 | 0.0 | 4.0 | 6.9 | | |
| Mastomys natalensis | 3.1 | 0.0 | 0.0 | 0.0 | 0.0 | 0.0 | 3.1 | 72.6 | |
| Otomys typus | 0.0 | 0.5 | 0.2 | 1.7 | 0.0 | 0.7 | 4.0 | 0 | 25.6 |
| Stenocephalemys albipes | 6.0 | 6.0 | 0.0 | 0.0 | 0.0 | 0.0 | 6.0 | 17.3 | |
| Stenocephalemys albocaudata | 0.0 | 0.0 | 0.2 | 6.0 | 3.3 | 3.3 | 17.1 | 0 | |
| Stenocephalemys griseicauda | 2.9 | 2.9 | 4.5 | 2.1 | 0.0 | 1.7 | 11.4 | 5.1 | |
| Tachyoryctes splendens | | | | | | | | | 30.5 |
| Crocidura baileyi | 0.0 | 0.5 | 0.0 | 0.0 | 0.0 | 0.5 | 4.3 | 0 | |
| Crocidura macmillani | 0.0 | 0.0 | 0.5 | 3.1 | 0.2 | 0.0 | 1.4 | 0 | |
| grand total | 12.1 | 2.14 | 11.4 | 29.5 | 17.4 | 27.4 | 100 | 0 | |

in Sherman live traps in the study. The Natal multimammate mouse, Ethiopian white-footed mouse and grey-tailed narrow-headed rat were all caught only in farmland (matrix). The remaining species were either captured in the buffer and core zones or in all three zones (table 3).

AWs took most of the rodents from *difit* (81% of events), followed by raptors (*Milvus migrans, Buteo augur*: 12%), caracals (*Caracal caracal*: 2.4%), domestic dogs (2.2%) and domestic cats (1.4%). EWs were observed taking rats in only 1% of *difit* scavenging events.

## 4. Discussion

This study provides the first detailed observational data on the foraging behaviour of AWs and provides inferences into the extent of dietary overlap with EWs. Earlier scat analyses indicated that rodents comprise a high proportion (47–57%) of the diet of the AW [8,9]. Here, we show that a large proportion (72%; electronic supplementary material, table S1) of the rodents consumed by AWs are obtained via scavenging from traditional traps (*difits*) rather than by hunting. Unlike the rodent specialist EWs [10,12], AWs feed on a greater diversity of food items, including insects, livestock carcasses and live sheep. Surprisingly, arthropods comprised the second most frequently consumed food items by AWs at 19.0%. Given that other sympatric mammals like EWs and gelada monkeys consume insects much less often [11,17], AWs may be unique among large mammals in the Ethiopian Highlands in exploiting a dietary niche in which insects play a major role.

AWs appear to be generally less efficient in capturing live rodents in Afro-alpine habitats (less than 17% success rate) than EWs, which exhibit capture efficiencies between 25 and 66% at Guassa [22] and 45% in the Bale mountains [10]. Whereas active hunting accounted for only 6% of the AW diet, EWs are almost exclusively (greater than 90%, [10]) rodent hunters and seldom scavenge (electronic supplementary material table S1). Thus, AWs exhibit a more omnivorous diet with a prominent scavenging component, whereas EWs are more strict rodent hunting specialists. This difference may be due to EWs preferring intact grassland habitat, and thus not encountering live rodents as frequently as carcasses. Further, the proclivity of AWs for scavenging rodents may reduce the extent of direct exploitative competition between AWs and EWs.

These results highlight the flexible nature of AW foraging behaviour. The foraging behaviour of AWs is highly seasonal and appears to track rodent abundance [8]. Consistent with previous research [8,9], we found that AWs forage on rodents more in the wet season and exploit livestock (sheep) more during the dry season. Indeed, while rodents comprise a large proportion of AW foraging efforts and capture

frequency, active hunting and scavenging of livestock was probably a major, if not the main, component of the diet in terms of biomass. AWs are also more proficient at rodent capture in farmlands compared to more intact habitats (bushland, grassland and woodland) where EWs thrive. The success of AWs in such disturbed habitats may be attributed to the higher visibility of farmland habitats, which evince little above-ground biomass, and/or the nature of rodent abundance and species composition in farmlands [23,24].

The reliance by AWs on rodents and insects implies that they play a role in pest control that may be beneficial to local farmers during certain times of the year [25]. Further, their scavenging of carcasses may have a hygiene benefit around human habitation [26]. Mesocarnivores seem to be on the increase in farm communities worldwide [27,28]. They benefit humans by feeding on crop vermin, and by removing garbage and carcasses, thus reducing health risks [26–28].

The present study points to two conservation recommendations that would facilitate the coexistence of EWs and AWs. First, given the adaptable nature of the foraging ecology of AWs in comparison to EWs, it is crucial that future EW conservation efforts focus on preserving intact habitats that are inherently preferred by EWs. Second, given the extent of persecution of AWs, local farmers should be informed about the potential benefits that AWs have for their farms.

# 5. Conclusion

This study shows that a large proportion of the rodents whose remains have been found in the scats of AWs were from dead animals caught in traditional traps, rather than obtained through predation, distinguishing them in their foraging habitats from EWs. As a consequence, we may conclude that exploitative food competition between the AW and EW is probably limited. This study also highlights the importance of AWs in rodent control (with their greater efficiency at capturing live rodents in farmland habitats) and waste management (through their removal of rodent and livestock carcasses near farms) in the Ethiopian Highlands. Lastly, it is important to underline that human agricultural expansion into EW habitats is likely attracting AWs, thereby adversely affecting EWs though interference competition [8].

Ethics. Ethiopian Wildlife Conservation Authority (EWCA): Policy for management of wildlife resources guidelines have been followed.

Data accessibility. Data available at the Dryad Digital Repository: https://doi.org/10.5061/dryad.p9g41sd [29].

Authors' contributions. G.T.M., A.A., A.B., C.S.Z. and N.C.S. conceived the study; G.T.M. and M.K. performed the fieldwork; G.T.M, A.A. and D.T. analysed the data and interpreted the results; G.T.M, A.A., V.V.V., P.J.F., D.T. and N.C.S. drafted the manuscript; and C.S.Z., J.M.R., V.V.V., M.K., A.B. and P.J.F. commented on the manuscript and contributed to its final version.

Competing interests. We declare we have no competing interests.

Funding. Rufford Small Grants Foundation, Mohamed Bin Zayed Species Conservation Fund, Jimma University and U.S.-Norway Fulbright Foundation.

Acknowledgement. We thank EWCA for research permission and the Ethiopian Wolf Conservation Programme for capture equipment and assistance capturing AWs.

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
