## [Reviewer comments · Royal Society Open Science]

Review History

RSOS-190772.R0 (Original submission)

Review form: Reviewer 1

Is the manuscript scientifically sound in its present form?

Yes

Are the interpretations and conclusions justified by the results?

Yes

Is the language acceptable?

Yes

Do you have any ethical concerns with this paper?

No

Have you any concerns about statistical analyses in this paper?

No

Recommendation?

Accept with minor revision (please list in comments)

Comments to the Author(s)

Gutema et al. Review

Foraging ecology of African wolves (*Canis lupaster*) and its implications for the conservation of Ethiopian wolves (*Canis simensis*)

July 2, 2019

Comments for authors:

This paper reports on foraging behavior of African wolves and potential competition with Ethiopian wolves. The authors mimic the methods of previous research on Ethiopian wolves for a direct comparison. They examine foraging behavior (hunting vs scavenging) in several habitats and prey selection. Their results have implications for Ethiopian wolf conservation as well as show the potential benefits of African wolves to farmers through reducing the rodent population and consuming dead rodents potentially reducing the spread of disease. They conclude that direct competition between African and Ethiopian wolves for food is limited but they highlight the importance of habitat protection in unmanaged (i.e., natural) landscapes.

This paper is generally well done but could use some clarification in the methods. Some of the methods are not well described and some methods (such as use of chi-squared) are not discussed at all but are presented in the results. I also encourage the authors to carefully proof-read their manuscript prior to submission as several of my comments reflect lack of attention to detail. Additionally, I think the real highlight of the paper is the potential benefits of African wolves and this should be more prominent in the paper. As well, the lack of food competition and what that means for Ethiopian wolves is an important conclusion.

Line 37: I don't understand what you mean by "totaling 24 rodents per hectare per day". Do you mean averaging? Or were there always 24 rodents /hectare/day captured by local traps? I also would include 80% (n=xx) so the reader doesn't have to do the math.

Line 38: carcasses – as in scavenged carcasses and not kills? I think this is what you mean.

Line 40: missing word here – anthropogenically altered what?

Line 62: endangered does not need to be capitalized

Line 64: I think this should be Afro-alpine

Line 65: add a comma after habitats

Line 66: change to: habitat loss and rabies and canine....

Line 69: you need to match tenses – so you say "scat analysis revealed (past tense) that the diet of AWs consists (present tense)...and varies (present tense)" probably best in past tense to reflect this was what the authors found at that time (and it may not hold true currently).

Line 72–76: So you know about the AWs diet from scat analyses, how do you know EWs are active rodent hunters and rarely scavenge? Were you actually watching EWs vs scat analyses for AWs? Make this clear.

Line 92–94: As I mentioned above, you should make it clear that this was done for EWs and now you are going to do it for AWs for direct comparison.

Figure 1: typo – core areax. I also think you should include a description of what the core area, buffer and matrix mean so the reader doesn't have to find this in the text. This is maybe a requirement of the journal anyway? Check the guidelines.

Line 105 and elsewhere: you should be using an en dash when you mean “to” and not a hyphen (e.g., 3,200–3,600m)

Line 106: typo? 153)

Line 112: What is “wet” and “dry”? This is subjective and means different things to different people. Please at least give some annual precipitation amounts and average high and low temperatures.

Line 123: missing closing parentheses)

Line 138–141: Probably need some latin names in here.

Table S1: Says sus = successful in caption but abbreviation in title is suc

Line 154: extra word here...visiting them in at....

Line 153–159: This is a bit confusing – the farmers set the traps and checked them every 1–4 hours. You checked them every 2 hours? And both you and the farmer would remove the carcass and reset the trap? How then, if you remove the carcass, do you account for this when counting how many rodents are eaten by carnivores and raptors?

Line 163–164: I think this should say mixed-effects (not mixed effect). And what are the localities – just the 2 (GMCCA and BSNP)?

Line 181: sometimes these sub-headings have been bold, sometimes not – check what is correct journal formatting. Also, I am not sure what this heading means. What is the accumulating part? Your headings should also match from section to section. So whatever you call it in your methods, it should also be called this in the results – and should always follow the same order to maintain flow.

Line 184: missing word here – resulting IN a total.....

Line 192: Is this a chi-squared result? Or what is X2? I didn't see this in the methods so it needs to be described in the methods.

Line 206: Now this answers my earlier question of whether the 24/hectare/day was an average. Please make that clear in your abstract as I mentioned above.

Table S2: This caption isn't very clear. I would use full sentences not phrases.

Line 207–211: Could be helpful to use common names here as well as latin names

Tables: Sometimes you use abbreviations (e.g., GMCCA) and sometimes you write it out. Check the journal guidelines – and be consistent with things like this in the manuscript.

Line 212: Incorrect use of the word only – it is modifying farmland not caught00they were not ONLY caught (they also live there, are seen there etc...) so.....were caught only in farmland (not were only caught).

Line 214: in the abstract you said 80% and now you say 81%--or maybe you are talking about something different here?

Line 216: typo – this should be taking rate from (not form). Also, you sometimes have difit, sometimes “Difit”, sometimes Difit. Pick one term to use and be consistent.

Line 234–235: are the habitats and conditions similar in these areas to those of your study area? And were these studies conducted during the same time of year/same season or under similar conditions?

Line 236: ref 10? I think you are referring to your citation here so you need to format this correctly.

Line 248: add comma after major, if not the main,

Line 249: Why is biomass underlined?

Or maybe there is an avoidance going on and EWs would take rodents from difit more often by they are displaced by AWs?

Table 2 and elsewhere: You need to define all of your abbreviations.

Review form: Reviewer 2

Is the manuscript scientifically sound in its present form?

Yes

Are the interpretations and conclusions justified by the results?

Yes

Is the language acceptable?

Yes

Do you have any ethical concerns with this paper?

No

Have you any concerns about statistical analyses in this paper?

No

Recommendation?

Accept with minor revision (please list in comments)

Comments to the Author(s)

This is a very interesting manuscript with some interesting findings related to the feeding habits of African wolves. The authors also discuss the potential implications African wolf feeding habits may have on the critically endangered Ethiopian wolf. With some relatively minor revisions, I believe this paper would make a valuable contribution to our understanding of the interactions between these two species and how man plays an important role in these interactions and the conservation of both species. Please see below for a list of minor comments (references to line numbers refer to the numbers on the inside):

- 1) line 40 (in abstract) after "altered" should include "landscapes" or "habitats" or something similar.
- 2) line 91-92: Might recommend more clearly defining the goal of the study. Seems like primary goal was to evaluate the foraging ecology of AWs in greater detail then to assess the potential effects this may have on EWs via competition for resources.
- 3) line 103: add "Park" after Borena-Saynt National
- 4) line 108: Are there any estimates of numbers of EWs in the 2 study areas?
- 5) lines 115-120 and 122-123: What is the significance of these "zones" to your analysis of the feeding ecology of AWs? Other than Figure 1 and Table 3, there is no discussion or reference about why "zones" are important. May consider removing all reference to zones from the manuscript or add a paragraph discussing what the significance of the zones are in relation to AW feeding ecology in the discussion.
- 6) line 122-123: was this done to obtain an actual estimate or simply to identify species presence and document frequency of occurrence (see lines 210-213)?? Also, might consider adding at least 1 additional sentence summarizing the methods used with the Sherman traps.
- 7) line 130-131: how were night observations made (spotlight, night-vision equipment, etc)?
- 8) line 135-137: definition of scavenging for study only included scavenging on rodents. What about scavenging on livestock carcasses?
- 9) line 137-141: might also include "sheep and other livestock" in the hunted prey description. Although the term "carcass" implies scavenging, when describing livestock carcasses and rodent carcasses, the authors might specify that these were scavenging events by placing the word "scavenge" in front of them.
- 10) line 159: How was this recorded (via direct observation, sign observed while checking traps, etc)?
- 11) line 184: "...resulting in a total of 2856 records." Describe exactly what these records were (foraging events, hunting events, scavenging events, etc.).
- 12) line 186: add "scavenging on" (or something similar) prior to livestock carcasses.
- 13) line 201-202: May consider also summarizing results from table S1 describing the number of feeding attempts related to scavenging of livestock carcasses in this paragraph? Seems important to include in results to demonstrate the prevalence of scavenging by AWs.

14) line 235-236: "Whereas active hunting accounted for only 6% of the AW diet,..." - does this reference "active hunting" of all food items (sheep, insects, etc.) or just rodents? Where did the 6% come from...this study or another study? I do not see reference to it in the results or any tables or figures. If it is from this study, may add that information to results section but if it is from another study, please provide the reference.

15) line 240: might consider adding "as frequently" after "carcasses."

16) line 256: consider adding "during certain times of the year" after "local farmers."

17) lines 255-259 and the second conservation recommendation in lines 264-266: Are there really enough AWs to provide a significant health benefit to humans "by removing garbage and carcasses?" How does this benefit AWs, and potentially EWs, in the long term? Considering the positives of AW foraging ecology to humans, and its' potential benefit for EWs, is great, but there seems to be several drawbacks associated with this line of thinking that were not discussed and should likely be included. First, farmer's may see this potential benefit for a portion of the year, but it might be hard for the farmer to see the benefit for the other half of the year when AWs are consuming the farmer's livestock. It will be challenging to convince folks of a benefit when there are still negative consequences (either perceived or real) associated with having an animal around. Also, if one is successful at convincing farmers of these benefits of AWs, it may have some unanticipated consequences for EWs and the farmers themselves. It seems that AWs are somewhat habituated to humans and human food resources (whether it be drift trapping, livestock, or garbage), as a result, humans may be partially responsible for maintaining higher numbers of AWs than might otherwise be expected in the area. Being highly reliant on anthropogenically modified landscapes and food resources may increase interactions between AWs and local dogs which has the potential to maintain high prevalence of disease in the highlands and possibly increase disease transmission risk to EWs from these sources. With AWs displaying some level of habituation to resources provided by humans, AWs will continue to hang around which likely increases the risk for negative interactions to occur between AWs and domestic animals through direct predation. When livelihoods are affected, it may be difficult to see the benefit of continuing to have the culprit around, but if the goal is to conserve AWs as well as EWs and show farmers the benefit of having wolves around, it might be more important to inform farmers about methods that might be used to reduce depredation risk first (such as grazing management practices, carcass management, aversive conditioning, etc). When livelihoods are less affected, then it tends to be easier to maintain an open-mind to the benefits of a particular species.

18) Table 3: EW column adds to 115.6%. Also, is the comparison of EW scat content from the same study area? If so, may explicitly state that.

Decision letter (RSOS-190772.R0)

29-Jul-2019

Dear Dr Stenseth

On behalf of the Editors, I am pleased to inform you that your Manuscript RSOS-190772 entitled "Foraging ecology of African wolves (*Canis lupaster*) and its implications for the conservation of Ethiopian wolves (*Canis simensis*)" has been accepted for publication in Royal Society Open

Science subject to minor revision in accordance with the referee suggestions. Please find the referees' comments at the end of this email.

The reviewers and handling editors have recommended publication, but also suggest some minor revisions to your manuscript. Therefore, I invite you to respond to the comments and revise your manuscript.

- Ethics statement

- Data accessibility

If you wish to submit your supporting data or code to Dryad (<http://datadryad.org/>), or modify your current submission to dryad, please use the following link:
<http://datadryad.org/submit?journalID=RSOS&manu=RSOS-190772>

- Competing interests

- Authors' contributions

- Acknowledgements

- Funding statement

Because the schedule for publication is very tight, it is a condition of publication that you submit the revised version of your manuscript before 07-Aug-2019. Please note that the revision deadline will expire at 00.00am on this date. If you do not think you will be able to meet this date please let me know immediately.

Supplementary files will be published alongside the paper on the journal website and posted on the online figshare repository (<https://rs.figshare.com/>). The heading and legend provided for each supplementary file during the submission process will be used to create the figshare page,

so please ensure these are accurate and informative so that your files can be found in searches. Files on figshare will be made available approximately one week before the accompanying article so that the supplementary material can be attributed a unique DOI.

on behalf of Kevin Padian (Subject Editor)
openscience@royalsociety.org

Associate Editor Comments to Author:

The reviewers consulted recommend that, with relatively minor modifications, your paper may be acceptable for publication: with this in mind, please ensure that you fully respond to (and incorporate) the recommended changes, as well as provide a point-by-point response to them with your revision. It would be helpful to include a tracked-changes version of the revision when you submit it. Congratulations, and we'll look forward to receiving your revision.

Reviewer comments to Author:
Reviewer: 1

Comments to the Author(s)
Gutema et al. Review
Foraging ecology of African wolves (*Canis lupaster*) and its implications for the conservation of Ethiopian wolves (*Canis simensis*)
July 2, 2019

Comments for authors:

This paper reports on foraging behavior of African wolves and potential competition with Ethiopian wolves. The authors mimic the methods of previous research on Ethiopian wolves for a direct comparison. They examine foraging behavior (hunting vs scavenging) in several habitats and prey selection. Their results have implications for Ethiopian wolf conservation as well as

show the potential benefits of African wolves to farmers though reducing the rodent population and consuming dead rodents potentially reducing the spread of disease. They conclude that direct competition between African and Ethiopian wolves for food is limited but they highlight the importance of habitat protection in unmanaged (i.e., natural) landscapes.

This paper is generally well done but could use some clarification in the methods. Some of the methods are not well described and some methods (such as use of chi-squared) are not discussed at all but are presented in the results. I also encourage the authors to carefully proof-read their manuscript prior to submission as several of my comments reflect lack of attention to detail. Additionally, I think the real highlight of the paper is the potential benefits of African wolves and this should be more prominent in the paper. As well, the lack of food competition and what that means for Ethiopian wolves is an important conclusion.

Line 37: I don't understand what you mean by "totaling 24 rodents per hectare per day". Do you mean averaging? Or were there always 24 rodents /hectare/day captured by local traps? I also would include 80% (n=xx) so the reader doesn't have to do the math.

Line 38: carcasses – as in scavenged carcasses and not kills? I think this is what you mean.

Line 40: missing word here – anthropogenically altered what?

Line 62: endangered does not need to be capitalized

Line 64: I think this should be Afro-alpine

Line 65: add a comma after habitats

Line 66: change to: habitat loss and rabies and canine....

Line 69: you need to match tenses – so you say "scat analysis revealed (past tense) that the diet of AWs consists (present tense)...and varies (present tense)" probably best in past tense to reflect this was what the authors found at that time (and it may not hold true currently).

Line 72-76: So you know about the AWs diet from scat analyses, how do you know EWs are active rodent hunters and rarely scavenge? Were you actually watching EWs vs scat analyses for AWs? Make this clear.

Line 92-94: As I mentioned above, you should make it clear that this was done for EWs and now you are going to do it for AWs for direct comparison.

Figure 1: typo – core areax. I also think you should include a description of what the core area, buffer and matrix mean so the reader doesn't have to find this in the text. This is maybe a requirement of the journal anyway? Check the guidelines.

Line 105 and elsewhere: you should be using an en dash when you mean "to" and not a hyphen (e.g., 3,200–3,600m)

Line 106: typo? 153)

Line 112: What is "wet" and "dry"? This is subjective and means different things to different people. Please at least give some annual precipitation amounts and average high and low temperatures.

Line 123: missing closing parentheses)

Line 138–141: Probably need some latin names in here.

Table S1: Says sus = successful in caption but abbreviation in title is suc

Line 154: extra word here...visiting them in at....

Line 153–159: This is a bit confusing – the farmers set the traps and checked them every 1–4 hours. You checked them every 2 hours? And both you and the farmer would remove the carcass and reset the trap? How then, if you remove the carcass, do you account for this when counting how many rodents are eaten by carnivores and raptors?

Line 163–164: I think this should say mixed-effects (not mixed effect). And what are the localities – just the 2 (GMCCA and BSNP)?

Line 181: sometimes these sub-headings have been bold, sometimes not – check what is correct journal formatting. Also, I am not sure what this heading means. What is the accumulating part? Your headings should also match from section to section. So whatever you call it in your methods, it should also be called this in the results – and should always follow the same order to maintain flow.

Line 184: missing word here – resulting IN a total.....

Line 192: Is this a chi-squared result? Or what is X²? I didn't see this in the methods so it needs to be described in the methods.

Line 206: Now this answers my earlier question of whether the 24/hectare/day was an average. Please make that clear in your abstract as I mentioned above.

Table S2: This caption isn't very clear. I would use full sentences not phrases.

Line 207–211: Could be helpful to use common names here as well as latin names

Tables: Sometimes you use abbreviations (e.g., GMCCA) and sometimes you write it out. Check the journal guidelines – and be consistent with things like this in the manuscript.

Line 212: Incorrect use of the word only – it is modifying farmland not caught⁰⁰they were not ONLY caught (they also live there, are seen there etc...) so.....were caught only in farmland (not were only caught).

Line 214: in the abstract you said 80% and now you say 81%--or maybe you are talking about something different here?

Line 216: typo – this should be taking rate from (not form). Also, you sometimes have difit, sometimes "Difit", sometimes Difit. Pick one term to use and be consistent.

Line 234–235: are the habitats and conditions similar in these areas to those of your study area? And were these studies conducted during the same time of year/same season or under similar conditions?

Line 236: ref 10? I think you are referring to your citation here so you need to format this correctly.

Line 248: add comma after major, if not the main,

Line 249: Why is biomass underlined?

Or maybe there is an avoidance going on and EWs would take rodents from difit more often by they are displaced by AWs?

Table 2 and elsewhere: You need to define all of your abbreviations.

Reviewer: 2

Comments to the Author(s)

This is a very interesting manuscript with some interesting findings related to the feeding habits of African wolves. The authors also discuss the potential implications African wolf feeding habits may have on the critically endangered Ethiopian wolf. With some relatively minor revisions, I believe this paper would make a valuable contribution to our understanding of the interactions between these two species and how man plays an important role in these interactions and the conservation of both species. Please see below for a list of minor comments (references to line numbers refer to the numbers on the inside):

- 1) line 40 (in abstract) after "altered" should include "landscapes" or "habitats" or something similar.
- 2) line 91-92: Might recommend more clearly defining the goal of the study. Seems like primary goal was to evaluate the foraging ecology of AWs in greater detail then to assess the potential effects this may have on EWs via competition for resources.
- 3) line 103: add "Park" after Borena-Saynt National
- 4) line 108: Are there any estimates of numbers of EWs in the 2 study areas?
- 5) lines 115-120 and 122-123: What is the significance of these "zones" to your analysis of the feeding ecology of AWs? Other than Figure 1 and Table 3, there is no discussion or reference about why "zones" are important. May consider removing all reference to zones from the manuscript or add a paragraph discussing what the significance of the zones are in relation to AW feeding ecology in the discussion.
- 6) line 122-123: was this done to obtain an actual estimate or simply to identify species presence and document frequency of occurrence (see lines 210-213)?? Also, might consider adding at least 1 additional sentence summarizing the methods used with the Sherman traps.
- 7) line 130-131: how were night observations made (spotlight, night-vision equipment, etc)?
- 8) line 135-137: definition of scavenging for study only included scavenging on rodents. What about scavenging on livestock carcasses?
- 9) line 137-141: might also include "sheep and other livestock" in the hunted prey description. Although the term "carcass" implies scavenging, when describing livestock carcasses and rodent

carcasses, the authors might specify that these were scavenging events by placing the word "scavenge" in front of them.

10) line 159: How was this recorded (via direct observation, sign observed while checking traps, etc)?

11) line 184: "...resulting in a total of 2856 records." Describe exactly what these records were (foraging events, hunting events, scavenging events, etc.).

12) line 186: add "scavenging on" (or something similar) prior to livestock carcasses.

13) line 201-202: May consider also summarizing results from table S1 describing the number of feeding attempts related to scavenging of livestock carcasses in this paragraph? Seems important to include in results to demonstrate the prevalence of scavenging by AWs.

14) line 235-236: "Whereas active hunting accounted for only 6% of the AW diet,..." - does this reference "active hunting" of all food items (sheep, insects, etc.) or just rodents? Where did the 6% come from...this study or another study? I do not see reference to it in the results or any tables or figures. If it is from this study, may add that information to results section but if it is from another study, please provide the reference.

15) line 240: might consider adding "as frequently" after "carcasses."

16) line 256: consider adding "during certain times of the year" after "local farmers."

17) lines 255-259 and the second conservation recommendation in lines 264-266: Are there really enough AWs to provide a significant health benefit to humans "by removing garbage and carcasses?" How does this benefit AWs, and potentially EWs, in the long term? Considering the positives of AW foraging ecology to humans, and its' potential benefit for EWs, is great, but there seems to be several drawbacks associated with this line of thinking that were not discussed and should likely be included. First, farmer's may see this potential benefit for a portion of the year, but it might be hard for the farmer to see the benefit for the other half of the year when AWs are consuming the farmer's livestock. It will be challenging to convince folks of a benefit when there are still negative consequences (either perceived or real) associated with having an animal around. Also, if one is successful at convincing farmers of these benefits of AWs, it may have some unanticipated consequences for EWs and the farmers themselves. It seems that AWs are somewhat habituated to humans and human food resources (whether it be drift trapping, livestock, or garbage), as a result, humans may be partially responsible for maintaining higher numbers of AWs than might otherwise be expected in the area. Being highly reliant on anthropogenically modified landscapes and food resources may increase interactions between AWs and local dogs which has the potential to maintain high prevalence of disease in the highlands and possibly increase disease transmission risk to EWs from these sources. With AWs displaying some level of habituation to resources provided by humans, AWs will continue to hang around which likely increases the risk for negative interactions to occur between AWs and domestic animals through direct predation. When livelihoods are affected, it may be difficult to see the benefit of continuing to have the culprit around, but if the goal is to conserve AWs as well as EWs and show farmers the benefit of having wolves around, it might be more important to inform farmers about methods that might be used to reduce depredation risk first (such as grazing management practices, carcass management, aversive conditioning, etc). When livelihoods are less affected, then it tends to be easier to maintain an open-mind to the benefits of a particular species.

18) Table 3: EW column adds to 115.6%. Also, is the comparison of EW scat content from the same study area? If so, may explicitly state that.

Author's Response to Decision Letter for (RSOS-190772.R0)

See Appendix A.

Decision letter (RSOS-190772.R1)

09-Aug-2019

Dear Dr Stenseth,

I am pleased to inform you that your manuscript entitled "Foraging ecology of African wolves (*Canis lupaster*) and its implications for the conservation of Ethiopian wolves (*Canis simensis*)" is now accepted for publication in Royal Society Open Science.

on behalf of Kevin Padian (Subject Editor)
openscience@royalsociety.org

Appendix A

Reviewer comments to Author:

Reviewer: 1

Comments to the Author(s)

Gutema et al. Review

Foraging ecology of African wolves (*Canis lupaster*) and its implications for the conservation of Ethiopian wolves (*Canis simensis*)

July 2, 2019

Comments for authors:

This paper reports on foraging behavior of African wolves and potential competition with Ethiopian wolves. The authors mimic the methods of previous research on Ethiopian wolves for a direct comparison. They examine foraging behavior (hunting vs scavenging) in several habitats and prey selection. Their results have implications for Ethiopian wolf conservation as well as show the potential benefits of African wolves to farmers through reducing the rodent population and consuming dead rodents potentially reducing the spread of disease. They conclude that direct competition between African and Ethiopian wolves for food is limited but they highlight the importance of habitat protection in unmanaged (i.e., natural) landscapes.

Our reply: Excellently summarized

This paper is generally well done but could use some clarification in the methods. Some of the methods are not well described and some methods (such as use of chi-squared) are not discussed at all but are presented in the results. I also encourage the authors to carefully proof-read their manuscript prior to submission as several of my comments reflect lack of attention to detail. Additionally, I think the real highlight of the paper is the potential benefits of African wolves and this should be more prominent in the paper. As well, the lack of food competition and what that means for Ethiopian wolves is an important conclusion.

Our reply: We found these suggestions important, and we include all the comments in the updated manuscript.

Line 37: I don't understand what you mean by "totaling 24 rodents per hectare per day". Do you mean averaging? Or were there always 24 rodents /hectare/day captured by local traps? I also would include 80% (n=xx) so the reader doesn't have to do the math.

Our reply: That is correct; on average 24.7 ± 8.5 rodents /hectare/day captured by local traps. We now make it clear as suggested. We also include n for the 80% in the updated version.

Line 38: carcasses—as in scavenged carcasses and not kills? I think this is what you mean.

Our reply: That is correct. We meant rodents not actively captured but found dead and eaten by African wolf

Line 40: missing word here—anthropogenically altered what?

Our reply: We meant anthropogenically altered landscape, we have now fixed this incomplete sentence

Line 62: endangered does not need to be capitalized

Our reply: Thanks. We now changed it so that it is not capitalized.

Line 64: I think this should be Afro-alpine

Our reply: We now spell Afro-alpine as suggested throughout the manuscript. (There is inconsistent spelling of this word in the literature.)

Line 65: add a comma after habitats

Our reply: We did as suggested.

Line 66: change to: habitat loss and rabies and canine....

Our reply: Thanks. We did as suggested.

Line 69: you need to match tenses—so you say “scat analysis revealed (past tense) that the diet of AWs consists (present tense)...and varies (present tense)” probably best in past tense to reflect this was what the authors found at that time (and it may not hold true currently).

Our reply: Thanks, we now present everything in the past tense form.

Line 72–76: So you know about the AWs diet from scat analyses, how do you know EWs are active rodent hunters and rarely scavenge? Were you actually watching EWs vs scat analyses for AWs? Make this clear.

Our reply: The Ethiopian wolf diet and foraging behaviour has been extensively studied by Sillero-Zubiri and Gottelli, 1995, Ashenafi et al., 2005, and others, which we now emphasize here. The African wolf diet however had never before been studied except for one study which was totally dependent on scat analysis.

Line 92–94: As I mentioned above, you should make it clear that this was done for EWs and now you are going to do it for AWs for direct comparison.

Our reply: Thanks, we now make it clear that we used published data on the diet and foraging behaviour of Ethiopian wolf in the comparisons with our study on the diet of African wolf.

Figure 1: typo—core areax. I also think you should include a description of what the core area, buffer and matrix mean so the reader doesn't have to find this in the text. This is maybe a requirement of the journal anyway? Check the guidelines.

Our reply: Thanks, we have fixed it.

Line 105 and elsewhere: you should be using an en dash when you mean “to” and not a hyphen (e.g., 3,200–3,600m)

Our reply: Thanks, we have fixed this issue.

Line 106: typo? 153)

Our reply: it has now been corrected.

Line 112: What is “wet” and “dry”? This is subjective and means different things to different people. Please at least give some annual precipitation amounts and average high and low temperatures.

Our reply: We now provide more details on the climates at the study sites.

Line 123: missing closing parentheses

Our reply: it is now corrected

Line 138–141: Probably need some latin names in her

Our reply: For small rodents, we could not identify them to species level as it is difficult to make the identification from a distance during observations of the wolves.

Table S1: Says sus = successful in caption but abbreviation in title is suc

Our reply: Thanks, we have now fixed it

Line 154: extra word here...visiting them in at....

Our reply: Thanks, we have now deleted the word in

Line 153–159: This is a bit confusing—the farmers set the traps and checked them every 1–4 hours. You checked them every 2 hours? And both you and the farmer would remove the carcass and reset the trap? How then, if you remove the carcass, do your account for this when counting how many rodents are eaten by carnivores and raptors?

Our reply: Thanks, we have now made this section clear. We checked the traps every 2 hrs, but the farmers checked the traps with different time interval between 1 and 4 hrs to reinstall traps which captured rodents. In two hours interval, we removed any rodent carcass not collected by wild animals to avoid recounting of rodents trapped in our study section.

Line 163–164: I think this should say mixed-effects (not mixed effect). And what are the localities—just the 2 (GMCCA and BSNP)?

Our reply: localities are used as fixed effect, and yes we used the two study sites as our intention is to compare the diet of the African wolf in the two areas before we sum them up for further analysis.

Line 181: sometimes these sub-headings have been bold, sometimes not—check what is correct journal formatting. Also, I am not sure what this heading means. What is the accumulating part? Your headings should also match from section to section. So whatever you call it in your methods, it should also be called this in the results—and should always follow the same order to maintain flow.

Our reply: We have now reviewed the manuscript and made changes to make the headings and subheadings uniform in style.

Line 184: missing word here—resulting IN a total.....

Our reply: We have done as suggested

Line 192: Is this a chi-squared result? Or what is X²? I didn't see this in the methods so it needs to be described in the methods.

Our reply: Sorry, this is by mistake. We have used logistic regression as pointed out in the method section and the value should be Z, and we have corrected it.

Line 206: Now this answers my earlier question of whether the 24/hectare/day was an average. Please make that clear in your abstract as I mentioned above.

Our reply: Thanks, we have now done this and modified the abstract accordingly

Table S2: This caption isn't very clear. I would use full sentences not phrases.

Our reply: We have now shortened the caption to one sentence as Twenty-five areas of farmlands assessed and rodents captured by difit per hectare

Line 207–211: Could be helpful to use common names here as well as latin names

Our reply: Thanks, we have now included the common name with all scientific names

Tables: Sometimes you use abbreviations (e.g., GMCCA) and sometimes you write it out. Check the journal guidelines—and be consistent with things like this in the manuscript.

Our reply: Thanks, we now have used the abbreviation though out

Line 212: Incorrect use of the word only—it is modifying farmland not caught00they were not ONLY caught (they also live there, are seen there etc...) so.....were caught only in farmland (not were only caught).

Our reply: Thanks, we have now fixed it.

Line 214: in the abstract you said 80% and now you say 81%--or maybe you are talking about something different here?

Our reply: Thanks, this is mistake in the abstract. It is 81%

Line 216: typo—this should be taking rate from (not form). Also, you sometimes have difit, sometimes “Difit”, sometimes Difit. Pick one term to use and be consistent.

Our reply: Thanks. We have now fixed it, it should be *difit*

Line 234–235: are the habitats and conditions similar in these areas to those of your study area? And were these studies conducted during the same time of year/same season or under similar conditions?

Our reply: The studies are yearly

Line 236: ref 10? I think you are referring to your citation here so you need to format this correctly.

Our reply: We have now fixed this

Line 248: add comma after major, if not the main,

Our reply: We did as suggested

Line 249: Why is biomass underlined?

Our reply: it is underlined by mistake, and we have now fixed it

Or maybe there is an avoidance going on and EWs would take rodents from difit more often by they are displaced by AWs?

Our reply: Your suggestion could be the case, but is difficult to test how the Ethiopian wolf range might be in the absence of African wolves. In our previous paper (8), we have showed that the Ethiopian wolf comes into contact with the African wolf in the buffer zone which probably restricts the Ethiopian wolf’s range through interference competition.

Table 2 and elsewhere: You need to define all of your abbreviations.

Our reply: Thanks, we have now done this.

Reviewer: 2

Comments to the Author(s)

This is a very interesting manuscript with some interesting findings related to the feeding habits of African wolves. The authors also discuss the potential implications African wolf feeding habits may have on the critically endangered Ethiopian wolf. With some relatively minor revisions, I believe this paper would make a valuable contribution to our understanding of the interactions between these two species and how man plays an important role in these interactions and the conservation of both species. Please see below for a list of minor comments (references to line numbers refer to the numbers on the inside):

1) line 40 (in abstract) after "altered" should include "landscapes" or "habitats" or something similar.

Our reply: That is right. We have now included landscape.

2) line 91-92: Might recommend more clearly defining the goal of the study. Seems like primary goal was to evaluate the foraging ecology of AWs in greater detail then to assess the potential effects this may have on EWs via competition for resources.

Our reply: We made the changes as suggested

3) line 103: add "Park" after Borena-Saynt Nationa

Our reply: We made the change

4) line 108: Are there any estimates of numbers of EWs in the 2 study areas?

Our reply: Yes, we have now included the population estimates for EWs at both localities

5) lines 115-120 and 122-123: What is the significance of these "zones" to your analysis of the feeding ecology of AWs? Other than Figure 1 and Table 3, there is no discussion or reference about why "zones" are important. May consider removing all reference to zones from the manuscript or add a paragraph discussing what the significance of the zones are in relation to AW feeding ecology in the discussion.

Our reply: Thanks, we now make clear that the different zones of the GMCCA are used by the African wolf and Ethiopian wolf to different degrees in the updated manuscript. In GMCCA, Ethiopian wolves use much of the central part of the protected area while African wolves use the buffer zone where much of the interference competition occurred at the border of core area and protected area.

6) line 122-123: was this done to obtain an actual estimate or simply to identify species presence and document frequency of occurrence (see lines 210-213)?? Also, might consider adding at least 1 additional sentence summarizing the methods used with the Sherman traps.

Our reply: Thanks, we found this sentence less important and we deleted it. In the rodent captures at 213 is about the rodents captured by the cultural traps, difit.

7) line 130-131: how were night observations made (spotlight, night-vision equipment, etc)?

Our reply: We used a spotlight but it is rarely difficult when there is much cloud cover

8) line 135-137: definition of scavenging for study only included scavenging on rodents. What about scavenging on livestock carcasses?

Our reply: Scavenging is for any carcasses including livestock carcasses. In table 1, we provide rodent and livestock carcasses separately.

9) line 137-141: might also include "sheep and other livestock" in the hunted prey description. Although the term "carcass" implies scavenging, when describing livestock carcasses and rodent carcasses, the authors might specify that these were scavenging events by placing the word "scavenge" in front of them.

Our reply: Thanks we have now provided this information

10) line 159: How was this recorded (via direct observation, sign observed while checking traps, etc)?

Our reply: Yes, this is from observation and we now include this in our updated manuscript.

11) line 184: "...resulting in a total of 2856 records." Describe exactly what these records were (foraging events, hunting events, scavenging events, etc.).

Our reply: Thanks we have now make this sentence clear. The records include the food item and whether it is successful or unsuccessful whenever there is hunting attempt.

12) line 186: add "scavenging on" (or something similar) prior to livestock carcasses.

Our reply: Thanks, we made the change

13) line 201-202: May consider also summarizing results from table S1 describing the number of feeding attempts related to scavenging of livestock carcasses in this paragraph? Seems important to include in results to demonstrate the prevalence of scavenging by AWs.

Our reply: Thanks. In table 1, we provide levels of scavenging for both livestock and rodents

14) line 235-236: "Whereas active hunting accounted for only 6% of the AW diet,..." - does this reference "active hunting" of all food items (sheep, insects, etc.) or just rodents? Where

did the 6% come from...this study or another study? I do not see reference to it in the results or any tables or figures. If it is from this study, may add that information to results section but if it is from another study, please provide the reference.

Our reply: We have provided this information in the supplementary material Table S1, and that is why you do not find it in the main document. We have now cited Table S1 at the end of this sentence to address this problem. Thanks for pointing it out.

15) line 240: might consider adding "as frequently" after "carcasses."

Our reply: We did as suggested

16) line 256: consider adding "during certain times of the year" after "local farmers."

Our reply: We made the change as suggested

17) lines 255-259 and the second conservation recommendation in lines 264-266: Are there really enough AWs to provide a significant health benefit to humans "by removing garbage and carcasses?" How does this benefit AWs, and potentially EWs, in the long term? Considering the positives of AW foraging ecology to humans, and its' potential benefit for EWs, is great, but there seems to be several drawbacks associated with this line of thinking that were not discussed and should likely be included. First, farmer's may see this potential benefit for a portion of the year, but it might be hard for the farmer to see the benefit for the other half of the year when AWs are consuming the farmer's livestock. It will be challenging to convince folks of a benefit when there are still negative consequences (either perceived or real) associated with having an animal around. Also, if one is successful at convincing farmers of these benefits of AWs, it may have some unanticipated consequences for EWs and the farmers themselves. It seems that AWs are somewhat habituated to humans and human food resources (whether it be drift trapping, livestock, or garbage), as a result, humans may be partially responsible for maintaining higher numbers of AWs than might otherwise be expected in the area. Being highly reliant on anthropogenically modified landscapes and food resources may increase interactions between AWs and local dogs which has the potential to maintain high prevalence of disease in the highlands and possibly increase disease transmission risk to EWs from these sources. With AWs displaying some level of habituation to resources provided by humans, AWs will continue to hang around which likely increases the risk for negative interactions to occur between AWs and domestic animals through direct predation. When livelihoods are affected, it may be difficult to see the benefit of continuing to have the culprit around, but if the goal is to conserve AWs as well as EWs and show farmers the benefit of having wolves around, it might be more important to inform farmers about methods that might be used to reduce depredation risk first (such as grazing management practices, carcass management, aversive conditioning, etc). When livelihoods are less affected, then it tends to be easier to maintain an open-mind to the benefits of a particular species.

Our reply: Thanks. Reviewer w here nicely articulates the entire conservation situation facing African wolves and Ethiopian wolves. We now include the sentence emphasising the human expansion towards the Ethiopian wolf habitat attracting African wolves which may

affect the existence of Ethiopian wolf through interference competition. This has been our main finding and recommendation in our previous study on the competition between African wolf and Ethiopian wolf so we cite that study here as well,

Gutema TM, Atickem A, Bekele A, Sillero-Zubiri C, Kasso M, Tsegaye D, Venkataraman VV, Fashing PJ, Zinner D, Stenseth NC. 2018a Competition between sympatric wolf taxa: an example involving African and Ethiopian wolves. Royal Society Open Science 5, 172207. (doi: 10.1098/rsos.172207)

18) Table 3: EW column adds to 115.6%. Also, is the comparison of EW scat content from the same study area? If so, may explicitly state that.

Our reply: That is correct. The EW column is frequency of occurrence of a given rodent species, and a given scat can have more than one rodent species leading the sum more than 100%. Yes, it is from the same study area and we now mentioned this clearly in the table legend of the updated version.